# TRACTABLE LARGE SCALE CALIBRATION WITH RL

## ABSTRACT

In this work we show that Reinforcement Learning (RL) is an effective algorithm for calibration problems at a scale which traditionally applied Bayesian approaches struggle. This work uses synthetic data, so has access to ground truth parameters and it can be seen that RL learns different, arguably better information for different parts of the learning process. These exciting results set the foundation for deeper consideration of RL in this space.

## 1 INTRODUCTION

The process of inferring latent values from available data is a commonly recurring challenge in science. For example, given the number of confirmed cases of COVID-19 in a country or even a city, one might need to determine the epidemiological parameters of the virus. This task, sometimes referred to as calibration or sometimes as history matching is an example of an inverse problem, and has a variety of commonly applied formulations (Bent et al., 2021; Saulnier et al., 2017). In many solutions to inverse problems, Bayesian methods feature strongly however we see an opportunity to consider a slightly different problem formulation, and to utilise information which is already present to overcome one of the challenges which will plague any Bayesian application, the curse of dimensionality. This is of specific importance for applications of these methods in contemporary contexts like the case of COVID-19 where we have years of data and would like to learn about the effects of interventions over time and the dynamics of the virus as it evolves and adapts. We are specifically interested in learning more than 20 parameters, a regime where Bayesian methods are known to struggle. In this work we explore the application of Reinforcement Learning and consider if the nature of the results which can be generated with it 1) are substantively distinct from Bayesian Optimisation, 2) can be used with the results from Bayesian Optimisation to generate better results as quantified by accuracy and uncertainty. The general problem being solved is the following: Given data from some phenomenon or object which we cannot see, and given access to domain insights encoded in a mathematical (or even a neural network) model which can simulate similar data, we would like to use the model, and the acquired data to infer properties or parameters describing the unseen phenomenon, thus solving the *inverse* problem. More detail on this problem can be found in Tarantola (2005). While this problem is straightforward to express, it is challenging for a variety of reasons. Of note, the observed data is rarely the same type of data predicted by the model, so there is an often lossy transformation performed on the data before it is collected. An example of this is case data being aggregated across multiple health facilities, or reported in batches at different times from when it was captured. These are natural issues that arise which cannot be avoided.

## 2 METHODOLOGY

Case data, the cumulative number of confirmed cases (daily) and the number of deaths for a period of N=280 days, were generated using a SIRD model, a type of compartmental model. This model derives the values for the number of individuals in a population in the Susceptible (S), Infected (I), Recovered (R) and Dead (D) compartments. These four numbers thus capture the state of the population over time. The rate at which individuals transfer between each of these compartments is encapsulated in deterministic parameterised equations. For this work, only one parameter, the daily transmission rates can be varied, and the interval in which they can change is an experimental variable. We consider two treatments, one where the transmission rates can change in 28 day intervals, and another in which they can change at 7 day intervals. Accordingly, in this work, we use 280 days of case data to infer the associated sequence of transmission parameters in one case learning

10 potentially unique values and the second case learning 40 potentially unique values. All other model parameters are fixed at their default levels.

The transmission rate, a continuous variable, is quantised into one of 500 equally distributed values (in future work we will consider algorithms which support continuous actions). In this work we compare a Bayesian Optimization algorithm implemented in Optuna[1] with an implementation of SARSA with a Fourier basis linear function approximation (Sutton & Barto, 2018). This RL algorithm was selected since it has generally good performance with a state space of dimension $k \leq 5$. To generate an inference of the parameter sequence likely to have created the considered data, 30 runs each of length 10, 000 episodes were executed for each algorithm. Barycenters were calculated for the resulting sets of data using three methods: the arithmetic mean across all sequences for each individual point in time (Petitjean et al., 2011), optimizing with Dynamic Time Warping (DTW) as a loss function (Schultz & Jain, 2018), and optimizing with a differentiable modification to the DTW Cuturi & Blondel (2017). The implentations for each of these is selected from tslearn[2].

## 3 RESULTS

From the images presented in Figure 1 it can be seen that the sequence of transmission parameters inferred through the use of RL in this work are different from the values inferred with Bayesian Optimization. Moreover, they both diverge from the ground truth during some of the 280 day duration, but they do so in different ways. For example, Bayesian optimisation has poor alignment with the ground truth data early in the sequence, while RL has poor alignment towards the end. Qualitatively, Bayesian Optimisation is unable to capture the peak in transmission rates at the beginning, a property which would be of great importance to the public health response. It is unclear why RL over estimates the data for the last 30 days, but this overestimation still results in better performance when quantified using Dynamic Time Warping as a measure (0.435 vs 0.568). In this work we characterise uncertainty by considering the variance in the results from the 30 independently generated inferences. In the regions where both methods diverge from the ground truth, the true value remains within the region of uncertainty. When increasing the temporal resolution to weekly transmission rates, the quadrupling the number of parameters to be inferred, the quantitative and qualitative performance of RL remains consistent. This is notable as Bayesian Optimization utilises more RAM, uses more time to generate the inferences and still is unable to outperform RL. Aggregating data from both methods does not result in a statistically significant increase in performance, but future work would explore conditions under which this would be possible.

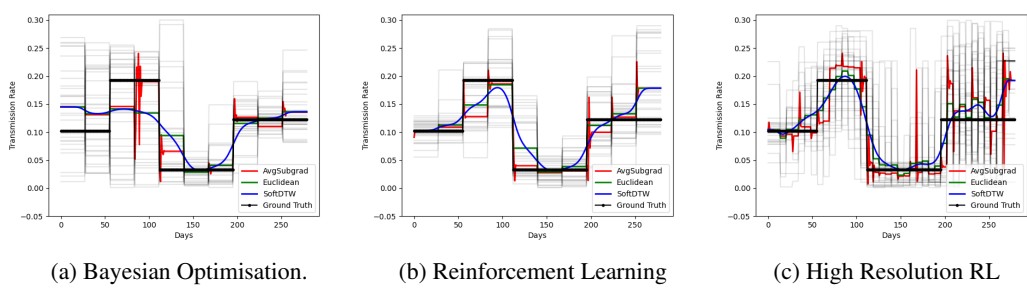

| (a) Bayesian Optimisation. | (b) Reinforcement Learning | (c) High Resolution RL |

Figure 1: Results

## 4 CONCLUSIONS

In this work the benefit of Reinforcement Learning in the solution of Inverse Problems with intermediate states are presented. This benefit is likely tied to the value of learning the system dynamics and the relationship between these dynamics and the resulting rewards. Future work is required to better characterise the benefit, and to better understand the contexts where this value will not be derived.

---

[1]https://optuna.readthedocs.io/
[2]https://tslearn.readthedocs.io/

## 5 URM STATEMENT

The authors acknowledge that each of the authors of this work meets the URM criteria of ICLR 2023 Tiny Papers Track.

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
