# OpenReview forum: "TRACTABLE LARGE SCALE CALIBRATION WITH RL"
_ICLR.cc/2023/TinyPapers — Submitted to Tiny Papers @ ICLR 2023_

### Official Review · Reviewer_SU7J · 2023-03-21

**Confidence:** 4

**Summary Of Contributions:**

This work suggests that reinforcement learning can be used as a competitive alternative to traditional Bayesian inference and system identification methods for estimating the time-varying parameters of a system.  A simple – yet relatively compelling – synthetic example using simulations from an SEIR model is presented.

**Rating:**

Great Start (GS): a submission which meets some of the reviewing criteria but has room for improvement

**Strengths And Weaknesses:**

I think this is a great start to a potentially really exciting idea: using reinforcement learning ideas to estimate the time-varying parameters of a simulator.  There is a rich history in system identification and Bayesian methods, and I believe this could be a new and very exciting approach to the problem.  I think there are also exciting aspects to it that the authors don’t highlight:
- (a) it places few (if any) restrictions on the simulator class,
- (b) parallelizable RL algorithms may be used to leverage parallel computation hardware, an area where Bayesian methods sometimes fall down,
- (c) RL algorithms can often use replay buffers further improving sample efficiency.
These should be highlighted, discussed, and celebrated!

However, I think the submission needs some work to make the proposed method clearer, highlight the problem it solves, highlight how the results show it does indeed solve it, and some more (qualitative) comparison to alternative methods.

## Clarity:
Overall the paper is a bit jumbled, and spends your very limited word count on things that aren’t super important.   For instance, most of the paragraph after “In this work we compare a Bayesian Optimization algorithm” could be dropped to a supplement, clearing space for more detailed discussion of the following:
- It is not immediately clear what “the problem” is here.  I believe it is a sequence prediction task, eg. given these observations, predict the sequence of parameters that best explain the data.  It would be good if the authors explicitly stated (with accompanying math) something like “the task is to estimate a sequence of T real-valued parameters from T observed data points.”  Then, later, for the SEIR example ”... these data points are the number of deceased individuals corrupted with i.i.d. Gaussian distributed noise”
- It is not clear how the authors “use RL” to solve the problem.  I believe you are suggesting that the agent learns to select the next parameter values given the observed data and the current transmission rate?  I recommend explicitly writing out the reward function that you use in the RL algorithm, what the policy is conditioned on, what it outputs and how it is constrained (R has to be positive, afterall) etc.
- It is not clear how I should interpret the results in Figure 1.  I don’t understand what the three coloured alternatives are.  Subgradients are not mentioned in the text, and so I don't understand what the red line represents.  Similarly, Euclidean and SoftDTW are not explained in the text.

## Correctness:
I think the method presented is _probably_ correct.  Unfortunately however there are just too few details for me to understand whether the proposed solution is correct.  A more detailed supplement would have been invaluable here.

## Reproducibility:
The same applies here as under *Correctness*.  I believe the results would be reproducible, but there is not enough information here to reproduce the results.

## Follows basic requirements:
The paper is suitably formatted.


**Suggested Changes:**

My overarching suggestion to the authors is to concretely state the problem, the metrics by which the solutions will be compared, your new proposed solution, and then the ways in which the proposed solution does/does not solve the problem compared to sensible baselines.  Use an appendix to present more details on the models, parameters, architectures, data etc, and clear space from the main text.

I would also like to see more comments on (although not necessarily evaluations of) some other competing methods.  For instance, you could use a message passing algorithm (assuming suitably defined likelihood functions exist), MCMC, or SMC in a random walk process on the transmission rate.  I would also like to see comments on how temporal smoothness could be enforced in each of the examples – the transmission rate is unlikely to “jump” overnight, and so encoding this prior information in how quickly R can change would dramatically improve both the Bayesian methods, and probably the RL method to (an L2 regularizer on the rate of change of the action).

There are a number of smaller changes that the authors could make as well to improve the paper:
- Footnotes should be avoided (and can also be formatted as citations to save some space).
- Fonts on figures should be roughly the same size as the body text.
- Only proper nouns should be capitalized (eg. “Bayesian Optimization” should be “Bayesian optimization”, “Dynamic Time Warping” -> “dynamic time warping”).
- The introduction is a very long paragraph.  It would be good to break it up into at least two paragraphs.
- Moreover, it would be good if you de-coupled your methodology and the example.  For instance, “Methodology” starts by explaining the model that you will apply it to – whereas your methodology is actually very general, and the SEIR model is just one example!  I would use a shorter introduction, have a “Motivating Example” section where you introduce the SEIR model, a shortened methodology section, and then the results.
- It is often better to move some discussion of results to the figure caption, as it can be difficult to flick backwards and forwards between the figure and the big block of text in Figure 3.  This can lead to important points being missed.

Good luck!  I look forward to seeing this presented somewhere!  :)

---

### Official Review · Reviewer_c6DQ · 2023-03-26

**Confidence:** 2

**Summary Of Contributions:**

This paper show that Reinforcement Learning (RL) is an effective algorithm for calibration problems at a scale which traditionally applied Bayesian approaches struggle

**Rating:**

Great Start (GS): a submission which meets some of the reviewing criteria but has room for improvement

**Strengths And Weaknesses:**

This paper proposed a promising and interesting problem by using RL in learning inverse problems (or calibration problems). The initial results seem promising.

Simultaneously, this paper could be improved by enhancing its clarity part. (see suggested changes for details)

**Suggested Changes:**

I would suggest authors.

- In Sec 1, it is better to use multiple paragraphs. The key message could be much more clear.
- In Sec 2, when presenting the proposed method, I suggest authors use a table (or figure) to present their methods clearly.
- In Fig 1, the caption could be better elaborated.

---

### Comment · Area_Chair_vwsd · 2023-06-06
**Check for Archival**

The authors have not submitted the revised version.

---

### Meta-Review · Area_Chair_vwsd · 2023-04-02

**Recommendation:** Invite to archive
**Confidence:** 4

**Metareview:**

This paper argues reinforcement learning (RL) is an effective technique to solve large-scale calibration problems, while the conventional Bayesian methods may struggle due to the curse of dimensionality. A simple synthetic example using simulations from an SEIR model is presented.

The motivation of this paper is clear, and the idea is promising. However, both reviewers point out the severe issue of presentation and clarity of this paper. The Reviewer `SU7J` also provides extensive constructive comments for the authors to further improve their work. A major revision is needed. More polishing of the writing, discussions, and experiments are needed before publication. The two main pages may not be sufficient, and hence the authors could add appendix.

Overall, based on the review criteria of the ICLR TinyPaper Track, the clarity and correctness of this paper need to be improved. Despite the limitations, it is a great start for the authors to practice writing research papers and further dive into the interesting research topic.

**Summary:**

This paper argues reinforcement learning (RL) is an effective technique to solve large-scale calibration problems, while the conventional Bayesian methods may struggle due to the curse of dimensionality. A simple synthetic example using simulations from an SEIR model is presented.

**Comments And Feedback To The Authors:**

Please carefully revise the paper following both reviewers' comments, especially the ones from Reviewer `SU7J`.

**Reason For Not Giving A Higher Recommendation:**

The clarity and correctness of this paper need to be improved. The current format does not meet the CCR standard.

**Reason For Not Giving A Lower Recommendation:**

The motivation of this paper is clear, and the idea is promising.

---

### Decision · Program_Chairs · 2023-04-10

Invite to archive